# A Zigzag but Upward Way to Develop an HIV-1 Vaccine

**DOI:** 10.3390/vaccines8030511

**Published:** 2020-09-08

**Authors:** Ziyu Wen, Caijun Sun

**Affiliations:** 1School of Public Health (Shenzhen), Sun Yat-sen University, Guangzhou 510006, China; wenzy3@mail2.sysu.edu.cn; 2Key Laboratory of Tropical Disease Control (Sun Yat-sen University), Ministry of Education, Guangzhou 511400, China

**Keywords:** HIV-1 vaccine, immune correlation, viral vector, protection

## Abstract

After decades of its epidemic, the human immunodeficiency virus type 1 (HIV-1) is still rampant worldwide. An effective vaccine is considered to be the ultimate strategy to control and prevent the spread of HIV-1. To date, hundreds of clinical trials for HIV-1 vaccines have been tested. However, there is no HIV-1 vaccine available yet, mostly because the immune correlates of protection against HIV-1 infection are not fully understood. Currently, a variety of recombinant viruses-vectored HIV-1 vaccine candidates are extensively studied as promising strategies to elicit the appropriate immune response to control HIV-1 infection. In this review, we summarize the current findings on the immunological parameters to predict the protective efficacy of HIV-1 vaccines, and highlight the latest advances on HIV-1 vaccines based on viral vectors.

## 1. A Brief History for HIV-1 Vaccine Development in Clinical Trials

At the nearly fourth decade of the human immunodeficiency virus type 1 (HIV-1) epidemic, UNAIDS (the Joint United Nations Program on HIV/AIDS), estimated that there were 37.9 million (32.7–44.0 million) people living with HIV-1, and 23.3 million people accessing antiretroviral therapy (ART). However, due to issues such as drug-resistant strains and the affordability of ART, an effective vaccine is still considered to be the optimal strategy to control and prevent the spread of the HIV-1 epidemic.

To date, hundreds of clinical trials for HIV-1 vaccines have been tested. The development process of HIV vaccines can be summarized in three stages: eliciting humoral immune responses (the first stage), eliciting T cell immune responses (the second stage), and the combined immunization that elicits both humoral and cellular responses (the third stage) (summarized in Table 1). The first stage was represented by two candidate vaccines (AIDSVAX B/B and AIDSVAX B/E), based on monomeric gp120 proteins, which were thought at that time to induce broadly neutralizing antibodies (bnAbs) against HIV-1 strains. However, the data of Phase III clinical trials indicated that neither of above candidate vaccines could provide protection against HIV-1 infection [1]. Nowadays, it is clear that monomeric gp120 proteins structurally are completely different from native trimeric Env proteins and are not able to induce bnAbs against primary HIV-1 strains. For a prophylactic vaccine, it is clearly having to induce bnAbs, whereas a cellular immune response is essential for development of a therapeutic vaccine. Therefore, attention was shifted to vaccines based on T-cell immunity. Viruses, such as adenoviruses and poxviruses, have been widely exploited as vectors to elicit cellular immune responses. One of the most remarkable results during the second stage was the human adenovirus 5 (Ad5)-based AIDS vaccine, though it was terminated due to no protection against HIV-1 infection during phase II b (STEP trial) [2]. After this failure, scientists have generally believed that an effective AIDS vaccine needed to elicit a balanced humoral and cellular immune response. Then, the phase III RV144 trial was conducted by a recombinant canarypox virus vector vaccine (ALVAC-HIV (vCP1521)) for initial immunization and a gp120 glycoprotein subunit vaccine (AIDSVAX B/E) for boosting immunization. The result showed that this strategy could reduce the risk of HIV infection in humans by 31.2% [3]. Although the interpretation for this protective efficacy was controversial among various researchers [4], there is no doubt that this partial protection helped to reignite the passion to develop an effective HIV vaccine. To confirm and optimize the protection efficiency of RV144, a large-scale phase IIb/III HVTN 702 trial was conducted in South Africa, and enrolled 5400 healthy volunteers who are sexually active between the ages of 18 and 35. Unfortunately, it was recently announced that the project HVTN702, known as “the only one close to success”, had been terminated in February 2020, due to no observed protection between vaccine recipients and placebos [5].

Recently, a promising strategy is the ongoing mosaic antigen-based HIV-1 vaccines. To solve the issue of genetic diversity of HIV-1 strains, the global HIV-1 sequences were analyzed by a genetic algorithm, and a kind of artificial mosaic antigen was designed to cover the largest scope of viral sequence diversity. Currently, two clinical trials of mosaic HIV-1 vaccine candidates, named Imbokodo (HVTN705) and Mosaico (HPX3002/HVTN 706), are in progress. For Imbokodo phase Ⅱ b trial, it consists of trivalent Ad26.Mos.HIV expressing mosaic HIV-1 Env/Gag/Pol antigens and clade C gp140 protein, and the result will be released in 2021. Mosaico trial is a complementary study to Imbokodo, and its composition includes not only clade C gp140 but mosaic gp140. Mosaico phase Ⅲ trial is being conducted at 55 sites throughout the United States, Latin America, and Europe, and its results will be released in 2023. 

## 2. A Way to Predict the Protective Efficacy of an HIV Vaccine Candidate in Humans: Immune Correlates of Protection

Although a great number of HIV-1 vaccine candidates have been explored, there is no HIV-1 vaccine available for clinic use yet, and we do not even fully understand which immune response(s) play a critical role in controlling HIV-1 infection [6,7]. To date, no immunoassay can exactly predict the protection efficacy of HIV-1 vaccine in humans, and this issue remains an important bottleneck in HIV vaccine development. Since this basic understanding is a prerequisite for the development of an effective HIV-1 vaccine, it is not surprising that a variety of HIV vaccine candidates have failed in clinical trials [2,8,9]. Before elucidating these immune correlates of protection, making an effective HIV-1 vaccine seems to be looking for a needle in a haystack. If there is no clue to evaluate the efficacy of a potential HIV-1 vaccine, how can we make an effective vaccine against HIV-1? In the past decades, increasing immunological parameters have been exploited to evaluate HIV vaccine candidates, and we herein summarize the updated parameters as shown in Figure 1.

### 2.1. Broad-Spectrum Neutralizing Antibodies

Traditionally, induction of neutralizing antibodies is the major goal for vaccine development. The HIV-1 genome is highly variable, so an effective HIV-1 vaccine requires to induce the broad-spectrum neutralizing antibodies (bnAbs) [10]. The recent termination of HVTN702 trial implied the importance of inducing HIV-1 bnAbs again. bnAbs can neutralize HIV-1 strains from diverse genetic and geographic backgrounds, mainly targeting some relatively conserved regions on the HIV envelope (Env) trimer, including the CD4 binding site (e.g., B12, VRC01, 3BNC60, 3BNC117), the variable regions 1, 2 and 3 (V1/V2/V3) region (e.g., PG9, PG16, CH01, PGT145) and surrounding glycans (e.g., 2G12, PGT121), gp120/gp41 interface (e.g., 35O22, 8ANC195), membrane-proximal external region (MPER) (e.g., 2F5, 4E10, Z13, 10E8), silent face center and fusion peptide (e.g., VRC34.01, ACS202) [11]. bNabs have some common features, such as polyreactivity for host antigens, extensive somatic hypermutation (SHM), and long variable heavy-chain third complementarity determining regions. Some bnAbs, including VRC01, 3BNC117, VRC07-523, N6, 10-1074, PGT121 10E8, PGDM1400, CAP256, etc., are being tested in clinical trials [12]. In parallel, it is still an ultimate goal for envelope-based vaccines to induce early and broad antibody responses which can neutralize conserved HIV-1 epitopes. However, an Env-based vaccine is a huge challenge yet, because the spatial configuration of Env protein is in dynamic changes with time. Several immunization strategies that elicit HIV-1 bnAbs have been reported, including adapting Env strains to induce specific neutralizing Ab lineages [13,14]; activating specific naive B cells for lineage expansion and maturation to induce desired lineages [15,16,17]; and an “epitope-based” immunization strategy against vulnerability sites on the Env trimer [18,19]. Some works had shown that SOSIP vaccines, a soluble recombinant HIV-1 envelope glycoprotein trimer close to native conformation, elicited robust autologous nAbs in rabbits and non-human primates (NHPs) [20,21], and the glycopeptide immunizations could successfully induce V3-glycan–targeted antibodies in rhesus macaques [18], and V2-SET-based immunogens induced V1V2-directed antibodies in guinea pigs [19]. Recent work showed that N-terminal residues of Env fusion peptide effectively elicited bnAbs in mice, which had neutralized up to 31% of a cross-clade panel of 208 HIV-1 strains [22]. Another work also showed that a fusion-peptide-primed vaccine regimen elicited a cross-clade HIV-1 neutralizing antibody in macaques with 59% neutralization breadth [23], implying the possibility to induce bnAbs by an HIV-1 vaccine.

### 2.2. Non-Neutralizing Antibodies

Besides bnAbs, it has been reported that non-neutralizing antibodies (nnAbs) also play important roles in controlling viral infection, including antibody-dependent complement deposition (ADCD) [24], antibody-dependent cellular cytotoxicity (ADCC) [25], antibody-dependent cellular phagocytosis (ADCP) [26,27], antibody-dependent cell-mediated viral inhibition (ADCVI) [25,28] and transactivator of transcription factor (Tat) -binding antibodies [29]. In particular, the detailed analysis showed that the partial protection of RV144 trial was correlated to non-neutralizing antibodies, rather than neutralizing antibodies. For example, the titers and affinity of plasma IgG ADCC binding antibody to gp120 V1/V2 in vaccinated volunteers were associated with reduced HIV-1 infection risk, but there was no detectable neutralizing antibody in these volunteers [30,31,32,33,34]. In general, the ADCC function is mediated by IgG Fc fragment, but recent studies have shown that those gp41-specific IgA antibodies are also involved in mediating ADCC activity [35]. Meanwhile, studies further demonstrated that IgA induces ADCP in a FcαRI-dependent manner [36]. Another study demonstrated that Ad26/MVA-vectored HIV-1 mosaic antigens achieved a significant protection against SHIV-SF162P3 challenges in rhesus monkeys, and this protection was associated with functional non-neutralizing antibodies and neutralizing antibodies, implying that the coordinated activity of multiple antibodies may be critical to control HIV-1 infection [37]. Therefore, the neutralizing antibody responses measured by standard assays in the peripheral blood may be incomprehensive for achieving a prophylactic vaccine-mediated full protection, and the non-neutralizing functional antibodies should also be evaluated.

### 2.3. Immune Breadth and Magnitude of Systemic Cellular Responses

HIV-1-specific cell-mediated immunity, particularly CD8^+^ T cells, has been recognized to contribute to the control of virus replication during chronic infection [38], and even reduce the threshold of nAbs to confer durable protection [39]. Currently, ex vivo interferon-gamma (IFN-γ)-mediated enzyme-linked immunospot (ELISPOT) assays have been widely adopted to measure the frequency of cellular responses induced by vaccination. However, some works have demonstrated that the immune breadth, compared with the magnitude of the cellular immune responses, might be more important in controlling HIV-1/SIV infection and replication [40,41]. Immune breadth represents the immune responses against multi-epitopes. One possible explanation for the failure of Merck’s STEP trial is that volunteers mounted limited and inadequate cellular-biased responses against only HIV-1 Gag, Pol, and Nef antigens [42,43]. The HIV-1 genome consists of 9 genes that encode 15 proteins; the immunogenicity of these proteins varies, and it is still not known which antigen-targeted immune responses would be more critical to control viral infection [44]. It was reported that a regimen of Ad5-vectored vaccine that consisted of all nine genes (*gag*, *pol*, *nef*, *vif*, *vpu*, *vpr*, *tat*, *rev*, and *env*) of the SIVmac239 as target antigens effectively elicited an immune response of balanced magnitude and enhanced breadth, and achieved effective protection in rhesus macaques [45]. Besides, an adenovirus vector encoding 18 CTL epitopes of HIV-1 was able to elicit broad, polyfunctional, and long-lived CD4^+^ T and CD8^+^ T cell responses [46]. However, recent work showed that the addition of Env component to a Gag/Pol-based vaccine led to reduced immune breadth and magnitude of Gag/Pol CD4^+^ T cellular responses [47]. Therefore, multi-antigen-based vaccines may offer a promising strategy for establishing broad-spectrum immunity and effective protection. Additionally, to reduce immune interference and competition of antigen components with the induction of the maximally achievable immune responses, the careful consideration of a minimalist set of antigens should also be required. 

### 2.4. Poly-Functional Lymphocytes

Although IFN-γ ELISPOT assay has been widely recognized to evaluate the cellular responses, there is growing evidence that this assay alone is insufficient to assess whether an HIV-1 vaccine would afford a protection [48]. Increasing data showed that immune protection might correlate with the frequency of a so-called poly-functional lymphocyte population, and these cells can specifically produce multiple cytokines and chemokines in addition to IFN-γ, including IL-2, TNFα, MIP-1, perforin and others [49,50,51,52]. In the STEP clinical trial, most vaccinated subjects (77%) produced a strong cellular immune response, as detected by IFN-γ ELISPOT assay, but ultimately impaired no protection [2,9]. Data analysis showed that 73% CD8^+^ T cells in those vaccines secreted IFN-γ alone and certain populations secreted TNF-α. However, only a few lymphocytes secreted IL-2 or multiple cytokines. A study pointed out that, in the RV144 clinical trial, HIV-specific CD4^+^ T-cell immune responses independently contribute to correlate of protection, which was mediated by polyfunctional effector memory CD4^+^ T cells which secreted various cytokines, such as IL-2, TNF-α, IFN-γ, IL-4 [53]. In addition, when compared with Ad5 vector immunization alone, prime/boost immunization via the different serotypes of adenovirus (such as Ad26/Ad5) induced stronger cellular immune responses. Importantly, more poly-functional CD8^+^ and CD4^+^ T lymphocytes secreting multiple cytokines (IFN-γ/TNFα/IL-2) were induced. After intravenous SIVmac251 virus challenge, the Ad26/Ad5 strategy afforded better control of viral replication [54,55]. Nevertheless, recent studies discovered that there was a subset of CD8^+^ T cell expressing chemokine receptor CXCR5, which exhibited a more potent proinflammatory function than CXCR5^−^ CD8^+^ T cells during chronic HIV-1 infection, because CXCR5^+^ CD8^+^ T cells enhanced the polyfunctionality of T cells and improved B-cell maturation [55,56,57]. Overall, induction of a polyfunctional CD8^+^ and CD4^+^ T lymphocytes biased immune responses should be considered as a priority for HIV-1 vaccine.

### 2.5. Mucosal Immunity

Sexual intercourse is the major transmission route of HIV-1 infection. Therefore, to develop an effective HIV-1 vaccine, it is critical to be capable of eliciting a robust mucosal immune response to block virus entry at mucosal sites. The secretory IgA antibody plays an important role in inhibiting HIV-1 mucosal transmission by blocking virus assembly and intracellular viral release [58]. Furthermore, effector memory T lymphocytes and specific cytotoxic T lymphocytes at mucosal sites can provide protection against a mucosal challenge [59]. In addition, protection can be provided by the innate immune cells at mucosal surface. For example, the monocyte, macrophage, dendritic cell (DC), NK cells, γδ T cells, innate lymphoid cell (ILC), innate-like lymphocyte (ILL), and others in the mucosal surface could produce cytokines, chemokines, and antiviral factors (such as APOBEC3G) [60,61]. The CC chemokines CCL-3, CCL-4, and CCL-5 were upregulated by immunization of SIVgp120 and gag p27, and then these chemokines bound and downmodulated CCR5, thereby inhibited HIV-1 entry into the host cells [62,63]. Another consideration in designing a mucosal vaccine is the selection of immunization routes. Different routes of immunization may result in various immune profiles. For example, administration of protein-based vaccines through the genitourinary tract could induce weak to modest local immune responses, but oral routes of immunization are less efficient in eliciting IgA in the mucosal surface of the vagina [64]. Interestingly, intranasal immunization has been shown to induce local immunity not only at mucosal sites in the respiratory tract but also in the genital tract [65]. Recently, an encouraging result was reported that immunization by intranasal routes with a vaccine comprised of gp41-subunit antigens achieved a good protection in challenged monkeys, and the activity of mucosal gp41-specific IgG and IgA was correlated with this protection [66]. However, disputes regarding the safety of intranasal immunization or immunization through the genitourinary tract must be resolved prior to application in humans.

### 2.6. T Cell Proliferation and Memory CD4^+^ T Cells

Studies demonstrated that HIV-1 specific CD4^+^ T and CD8^+^ T cells in the long-term non-progressive HIV-1 patients had strong ex vivo proliferation capacities, whereas this function was significantly decreased in chronically progressive HIV-1 patients [67,68]. More importantly, there was a positive correlation between cell proliferation capacity with the quantity of CD4^+^ T cells, but an inverse correlation between the proliferation capacity with the plasma viral load in HIV-1 patients [69]. Consequently, the induction and maintenance of proliferation of specific T lymphocytes have becoming an important immunological parameter to evaluate an effective HIV vaccine candidate. CFSE staining and Ki67-positive assay have been used widely for the analysis of cell proliferation. 

Memory CD4^+^ T cells, including central memory and effector memory CD4^+^ T cells in the intestinal mucosa, are the main target of HIV-1 [70]. During acute HIV-1 infection, a massive loss of memory CD4^+^ T cells throughout the body occurs, particularly in the mucosa. Therefore, the evaluation of a potential vaccine efficacy must include its ability to prevent destruction of memory CD4^+^ T cells.

## 3. The Current Status of HIV Vaccines Based on Viral Vectors

One of key prerequisites to develop an effective vaccine is to choose an appropriate delivery system, which can carry the target antigen gene into host cells to stimulate immune responses. Recombinant viruses have been extensively exploited as vectors for vaccines, and currently a variety of viral-based vectors, including adenovirus, poxvirus, cytomegalovirus (CMV), Sendai virus and varicella stomatitis virus (VSV), etc., have being developed as vectors for HIV-1 vaccines (summarized in Table 2). In general, the different viral vectors can elicit various profiles of immune responses, which contribute to a different role in protection efficacy. For example, one study showed that an ALVAC-based vector preferentially induced CD8^+^ T cell proliferation, while Ad5-based vector induced CD4^+^ T cell proliferation [71]. Here, we discuss the progress of promising viral vector-based HIV-1 vaccines, and mainly focus on adenovirus, poxvirus, CMV, and Sendai virus-based HIV-1 vaccines.

### 3.1. Adenovirus-Vectored HIV-1 Vaccines

Recombinant adenoviruses, especially replication-defective Ad5, have been extensively exploited as vectors for gene therapy or vaccines against HIV-1, hepatitis virus, and other infectious diseases [85,86,87]. However, a lack of protective efficacy against HIV-1 infection among a small subset of uncircumcised recipients with high levels of preexisting Ad5-neutralizing antibodies was observed after immunization with the Ad5-based HIV vaccine in the STEP trial. Some reports showed that the vaccination of Ad5-seropositive subjects may potentially create extra targets for HIV-1 infection [88,89]. One study also revealed that Ad5-specific CD4^+^ T cells were much more susceptible to HIV-1 infection than CMV-specific CD4^+^ T cells [90], and it might be related to the upregulated α4β7 integrin expression on CD4^+^ T cells. The α4β7 integrin may cause CD4^+^ T cells to migrate into the gastrointestinal tissue [91], where CD4^+^ T cells are mainly depleted during acute HIV-1 infection [92]. To overcome the issue of preexisting anti-Ad5 immunity due to natural infection, some works indicated that shielding Ad5 capsids by polyethylene glycol (PEG) or other polymers may bypass the detrimental effect of preexisting anti-adenovirus immunity [93,94]. To note, a series of Ad serotypes from human and other species have been explored as an alternative vector for developing HIV-1 vaccines [95,96]. For example, Ad26 and Ad35 vectors have been tested in clinical studies [97,98,99]. Inspiringly, phase I/II trials showed that priming with Ad26 vector expressing mosaic HIV-1 Env/Gag/Pol antigens (Ad26.Mos.HIV) and then boosting with Ad26.Mos.HIV plus high-dose clade C Env gp140 protein had elicited strong Env-specific binding antibody responses (100%), ADCP responses (80%) and T-cell responses (83%) in humans. Meanwhile, it also has induced similar magnitude, durability, and phenotype of immune responses in rhesus monkeys, which afforded 67% protection against SHIV-SF162P3 infection [100]. Compared with Ad5-based vaccine, Ad26-based vaccine has better protection efficacy against SIVmac251 challenge [101,102], and it does not increase the number and activation status of CD4^+^ T cells on human mucosal surface after vaccination. Furthermore, adenovirus vectors have been studied in combination with several other vectors, including DNA vector [101], CMV vector [103], MVA vector [104], MVTT vector [105], and other vectors [54], which elicited the augmented magnitude, breadth and polyfunctionality of cellular immune responses. Therefore, the next-generation Ad-vectored HIV-1 vaccines will be developed as a promising component of heterologous prime/boost strategies.

### 3.2. Poxvirus-Vectored HIV-1 Vaccines

A variety of poxviruses have been developed as gene delivery vector, including the modified vaccinia virus Ankara (MVA) strain, the New York vaccinia virus (NYVAC) strain, vaccinia virus TianTan (VTT), the canarypox strain (ALVAC), etc. Currently, HIV-1 vaccines based on MVA [106], NYVAC [75], and VTT [77] vectors have been clinically tested. However, the immunogenicity of these vectors may be attenuated due to pre-existing immunity in persons vaccinated against smallpox. Recently, researchers have taken a variety of measures to improve their immunogenicity [107]. For example, the vaccinia virus-based vectors are often used in combination with heterologous vectors, such as recombinant influenza viruses, adenoviruses [108], DNA vectors [109,110], VSV [111], or other poxvirus strains [112]. Specifically, a recent clinical trial revealed that priming immunization with NYVAC vectors and boosting with Env protein induced a high level of V1/V2 IgG antibody responses [75]. In addition, there are other strategies to enhance the immunogenicity, including genetically engineering of immunoregulatory genes [113,114,115], optimization of poxvirus promoters [116], co-expressing the immunostimulatory molecules [117], and administration adjuvants [118]. Reports revealed that ALVAC cause higher levels of pro-inflammatory and interferon-related cytokines and chemokines after immunization than its counterpart of MVA and NYVAC [119], and MVA preferentially causes CD8^+^ T cell responses, while NYVAC causes more CD4^+^ T cell responses [120,121].

### 3.3. CMV-Vectored HIV-1 Vaccines

Compared to Ad and Poxvirus-based vectors, cytomegalovirus (CMV)-based vectors can persist in the vaccinated individuals, and maintain the high-frequency of both circulating and tissue-resident effector memory T cell responses. For example, the immunization with rhesus CMV68.1 (RhCMV68.1) carrying SIV antigen inserts protected more than 50% of monkeys against the highly virulent SIV challenge [103]. The tissues from those protected monkeys did not show any residual SIV RNA or DNA by ultrasensitive PCR analysis [122]. Furthermore, one study revealed that RhCMV vectors elicited an unconventional CD8^+^ T cell population, which recognized antigen peptides either in the context of MHC-II or the non-classical, highly conserved MHC-E molecule [123,124]. CMV strains in different hosts are highly species-specific, and some works have shown that similar CD8^+^ T cell response profiles might be elicited by designing genetically modified human CMV vectors [125]. In addition, the attenuated RhCMV-vectored vaccine was highly efficacious against intravaginal SIVmac239 challenge [126].

### 3.4. Sendai-Vectored HIV-1 Vaccines

Sendai virus (SeV) is a negative-strand RNA virus, and it has been considered to be a safe vector for gene delivery, because it cannot integrate into the host genome. Although studies have indicated that antibodies against human parainfluenza virus type 1 (hPIV-1) can cross-react with SeV, SeV-specific antibodies in humans are not highly prevalent [127], implying that SeV-based vector might avoid pre-existing immune responses in general population to a certain extent. Some studies have shown that both replication-competent SeV vectors and replication-deficient SeV vectors carrying SIV Gag antigen (SeV-Gag) induced SIV Gag-specific CTL responses, thereby effectively controlling pathogenic SHIV89.6P challenge [128]. To date, SeV-based HIV-1 vaccine has been launched in phase I trials. A clinical trial, priming intranasally with replication-competent SeV vector expressing HIV-1 Gag and then boosting with Ad35-vectored HIV-1 vaccines, elicited a robust and long-lasting HIV-specific T cell responses and antibody responses [81].

## 4. Concluding Remarks and Perspective

As one of the most challenging scientific issues, the development of an HIV-1 vaccine is urgently desired but full of hardships. From the initial antibody-based strategy, then the T cell-based strategy, the heterologous prime/boost regimen, and up to the ongoing “mosaic antigen” strategy, the path of HIV-1 vaccine research is twisting and turning. To go through the dilemmas in HIV-1 vaccine development, researchers are making the unremitting efforts in the process of major scientific issues, such as novel immunogen design, vaccine delivery system, immunization routine, evaluation models, etc. 

Currently, the most important consensus in this field is that an effective HIV vaccine should be able to elicit: (1) high level of broad-spectrum neutralizing antibodies; (2) augmented magnitude, breadth and polyfunctional of cellular immune responses; (3) balanced immunity between humoral immune responses and cellular immune responses; (4) powerful and persistent immune responses at mucosal surfaces. The next generation of HIV vaccine candidates with these characteristics would confer better immune protection against HIV-1 infections. Among them, recombinant viruses-based vectors remain an attractive delivery vehicle for HIV-1 vaccine. In particular, HIV-1 vaccine should be further studied in combination with heterogeneous prime and boost strategies. Although nearly 40 years have passed since the discovery of AIDS, there is still no effective HIV vaccine available. The road to thoroughly eliminate the HIV pandemic will be covered of thistles and thorns, but we believe that a safe and protective HIV vaccine would become available within not so-far future.

## Figures and Tables

**Figure 1 vaccines-08-00511-f001:**
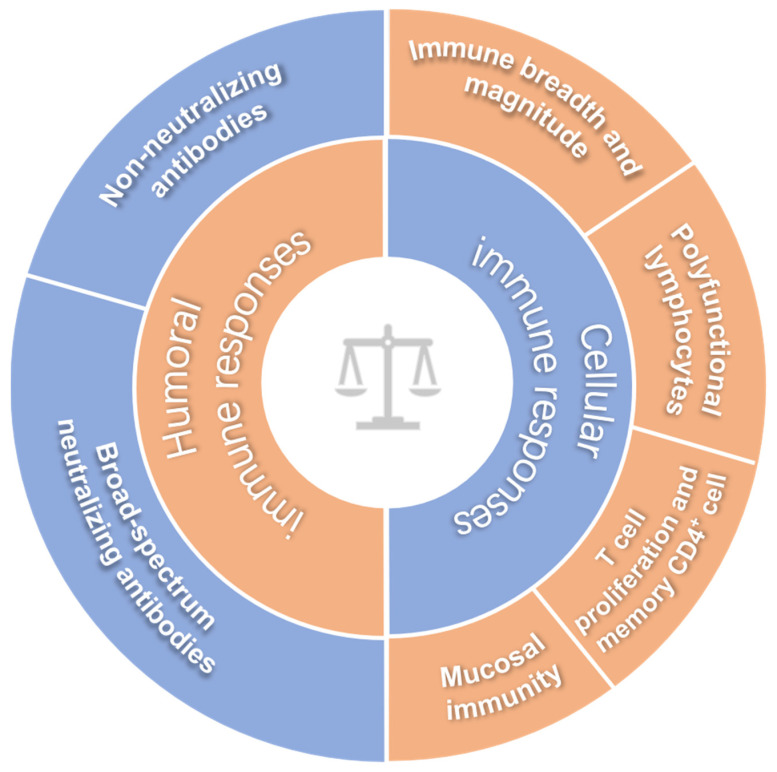
The inferred immune correlates of protection for HIV-1 vaccines.

**Table 1 vaccines-08-00511-t001:** Landmark of clinical trials for the HIV vaccines ^1^.

Phase	Trial ID	Strategy	Candidate	Volunteers	Outcome
III	VAX 003	Protein	AIDSVAX B/E	2500	No protection
III	VAX 004	Protein	AIDSVAX B/B	5400	No protection
IIb	HVTN 502/Merck 023 (STEP Study)	Viral Vector—Adeno	MRKAd5 HIV-1 gag/pol/nef	3000	Terminated because of no protection
IIb	HVTN 503 (Phambili)	Viral Vector—Adeno	MRKAd5 HIV-1 gag/pol/nef	3000	Terminated because of no protection
III	RV 144	Viral Vector—Pox/Protein	ALVAC-HIV-1(vCP1521)/AIDSVAX B/E	16403	Partial protection
III	HVTN702	Viral Vector—Pox/Protein	ALVAC-HIV/AIDSVAX B/E	5400	Terminated because of no protection
IIb	Imbokodo (HPX2008/HVTN705)	Viral Vector—Adeno/Protein	Ad 26-moasic/gp140	2637	The data will be released in 2021
III	Mosaico (HPX3002/HVTN706)	Viral Vector—Adeno/Protein	Ad 26-moasic/gp140	3800	The data will be released in 2023

^1^ Summarized and referenced from the IAVI trials database (http://www.iavireport.org/trials-db/Pages/default.aspx) as of June 2020.

**Table 2 vaccines-08-00511-t002:** Summary of promising viral vectored HIV-1 vaccines.

Viruses	Vectors	Vaccine Ingredients	Progress	Outcomes	Ref
Adenovirus	Ad5	MRKAd5 HIV-1 gag/pol/nef	In phase II trial	Terminated because of no protection	[2]
	Ad26	Ad 26-mosaic/gp140	In phase III trial	The data will be released in 2023	[72]
Poxvirus	MVA	DNA prime/MVA-HIV-1 boost	In phase I trial	Cellular and humoral immune responses	[73,74]
	NYVAC	DNA and gp120 prime/NYVAC-HIV and gp120 boost	In phase I trial	Env V1/V2 IgG binding antibody responses	[75]
	ALVAC	ALVAC-HIV (vcp1521) prime/AIDSVAX B/E boost	In phase III trial	Partial protection (31.2%)	[3]
	flow pox	DNA prime/rFPV–HIV gag, pol boost	In phase I trial	No protection	[76]
	Tiantan	DNAprime/recombinant attenuated vaccinia virus Tian Tan-gag, pol, env, nef boost	In phase II trial	In progress	[77]
Herpes virus	CMV	RhCMV-SIV Gag, Env, rev/tat/nef	preclinical	Control infection	[78]
	VZV	VZV-SIV env, gag	preclinical	Neutralizing antibodies and cellular immune responses	[79]
Flavivirus	flavivirus vector	DNA prime/flavivirus vector-env or gag-pol-nef boost; NYVAC-env or gag-pol-nef prime/flavivirus vector-env or gag-pol-nef boost	preclinical	IgG, ADCC and T cell responses	[80]
Paramyxovirus	sendai virus	SeV-Gag prime/Ad35-GRIN boost	In phase II trial	Functional, durable HIV-specific T-cell responses and boosted antibody responses	[81]
	measles virus	MV1-F4-HIV-1 fusion protein	preclinical	No shedding of infectious viral particles was identified	[82]
Rhabdovirus	rabies virus	RV-SHIV89.6P env and SIVmac239 gag	preclinical	Control infection	[83]
	VSV	rVSV HIV-1 gag	In phase I trial	Gag-specific T-cell responses were detected in 63% of participants	[84]

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
