# Peer review of "A Zigzag but Upward Way to Develop an HIV-1 Vaccine"

_vaccines, 2020, doi:10.3390/vaccines8030511_

Round 1

Reviewer 1 Report

For the Authors:

The authors briefly describe the history of HIV vaccine development and move the question of what immune responses correlates with and are key players in protection. They include broad spectrum neutralizing Ab, non-neutralizing Ab, immune breadth (induced by continuous stimulation with multi-epitope Ag), polyfunctional T cell population (producing various cytokines), mucosal immunity, etc. These topics are divided and described in subsections and are concise and easy to read. There are lots of information we can learn from this section. Characteristics of this review are that they discuss the immune correlates of protection but do not merely introduce the results of human clinical trials. Of course, the authors introduce the past and current human clinical studies as well as non-human primate studies but readably explain what we learnt about from each clinical study/trial. They also introduce the current promising clinical studies, Imbokodo and Mosaico trials, ongoing with Ad26 virus containing HIV mosaic Ag.

Current HIV vaccine candidates are mostly virus vector-based vaccines, such as adenovirus and poxvirus. This review well describes the pros and cons of each virus vector, namely the efficacious immune responses and adverse responses induced by each virus vector, and they are easy to understand. This point is also rarely seen in other reviews.

Overall, this review is concise but well-written and easy to read.

Minor points (need to be amended):

1) Line168: --- CD4+ and CD8+ T cells expressing some immune ---.

2) Line 237: Please delete “Results” from the section title.

3) Line 284: Please delete “virus”.

Author Response

The authors briefly describe the history of HIV vaccine development and move the question of what immune responses correlates with and are key players in protection. They include broad spectrum neutralizing Ab, non-neutralizing Ab, immune breadth (induced by continuous stimulation with multi-epitope Ag), polyfunctional T cell population (producing various cytokines), mucosal immunity, etc. These topics are divided and described in subsections and are concise and easy to read. There are lots of information we can learn from this section. Characteristics of this review are that they discuss the immune correlates of protection but do not merely introduce the results of human clinical trials. Of course, the authors introduce the past and current human clinical studies as well as non-human primate studies but readably explain what we learnt about from each clinical study/trial. They also introduce the current promising clinical studies, Imbokodo and Mosaico trials, ongoing with Ad26 virus containing HIV mosaic Ag.

Current HIV vaccine candidates are mostly virus vector-based vaccines, such as adenovirus and poxvirus. This review well describes the pros and cons of each virus vector, namely the efficacious immune responses and adverse responses induced by each virus vector, and they are easy to understand. This point is also rarely seen in other reviews.

Overall, this review is concise but well-written and easy to read.

Answer: Thanks for your kindly comments.

Minor points (need to be amended):

  • Line168: --- CD4+ and CD8+ T cells expressing some immune ---.

Answer: We are sorry for this mistake, and we have corrected it in our revised manuscript. Thank you.

  • Line 237: Please delete “Results” from the section title.

Answer: We are very sorry for this abundant word when we adapted the template. We have deleted this word. Thank you.

3) Line 284: Please delete “virus”.

Answer: Thanks for you careful reading, and we have deleted this abundant word.

Reviewer 2 Report

Several decades after the discovery of HIV as a causal agent of AIDS, we still do not know certainly which immune responses are needed to block or eliminate the virus. However, over the last years great progress has been made to define those immune parameters that are related to some degree of protection in both preclinical and clinical trials. This topic has been thoroughly reviewed over the past few years for their relevance in the vaccine development field.

This manuscript does not update or extend the results that have been previously discussed in other reviews dealing with this topic. There are misleading concepts and the English language and style require extensive editing. The authors perform an incomplete analysis of the clinical trials that used viral vectors as HIV vaccines.

Author Response

Several decades after the discovery of HIV as a causal agent of AIDS, we still do not know certainly which immune responses are needed to block or eliminate the virus. However, over the last years great progress has been made to define those immune parameters that are related to some degree of protection in both preclinical and clinical trials. This topic has been thoroughly reviewed over the past few years for their relevance in the vaccine development field.

This manuscript does not update or extend the results that have been previously discussed in other reviews dealing with this topic. There are misleading concepts and the English language and style require extensive editing. The authors perform an incomplete analysis of the clinical trials that used viral vectors as HIV vaccines.

Answer: Thanks for your careful and strict review for our manuscript. As you mentioned, studies about HIV vaccine development has been an “old” topic more than three decades, so this topic has been often reviewed over the past years. But we do not agree what you said --- “this topic has been thoroughly reviewed”. Actually, just as you mentioned, “we still do not know certainly which immune responses are needed to block or eliminate the virus”, this field is still one of the most challenging scientific issues nowadays, and there are increasing new research and exploration in this field. In fact, our manuscript has included the results of the latest AIDS vaccine research, such as mosaic antigen-based clinical trials (Imbokodo, Mosaico, and HVTN702, etc.). Moreover, we firstly summarize the current findings on the immunological parameters to predict the protective efficacy of HIV-1 vaccines, and we do not think this summary has been covered in other published HIV vaccine- related literatures. We think our work would be helpful to provide implications to design the novel vaccine strategies against HIV-1 infection. 

With regard to “an incomplete analysis of the clinical trials that used viral vectors as HIV vaccines”, we mainly discussed the adenovirus, poxvirus, CMV, and Sendai virus-based HIV vaccines in the main text. But we also listed in detail the most promising viral vectors-based HIV vaccines in completed preclinical and/or clinical stages, such as flavivirus, VZV, measles virus, rabies virus, VSV etc. For this information, please refer to Table 2 in our manuscript.

As for “misleading concepts and the English language and style”, we are sorry for these mistakes and not perfect English language, and the revised manuscript has been carefully edited by a native English speaker. After extensive revisions, we believe that this review has been substantially improved. Thanks for your understanding.

Reviewer 3 Report

Recommendation: Publish after minor revision

In this review article the authors provide a comprehensive review of current HIV-1 vaccine strategies, their status and clinical outcome. Then, the authors provide a detailed overview of viral vector based HIV-1 vaccine strategies. In my opinion, this review article is nicely written, contain a concise and detail information of HIV-1 vaccine strategies.

Minor Comments:

The title of heading 3 need to be re-formatted.

Author Response

In this review article the authors provide a comprehensive review of current HIV-1 vaccine strategies, their status and clinical outcome. Then, the authors provide a detailed overview of viral vector based HIV-1 vaccine strategies. In my opinion, this review article is nicely written, contain a concise and detail information of HIV-1 vaccine strategies.

Answer: Thanks for your kindly comments.

Minor Comments:

The title of heading 3 need to be re-formatted.

Answer: We are very sorry for this mistake. We have deleted the abundant word “Results” in this heading. Thank you.

Round 2

Reviewer 2 Report

The review has been substantially improved after an extensive revision and inclusion of updated references.

Minor points:

Line 91-97: The authors are mentioning examples of bnAbs targeting different vulnerability sites on the Env trimer. It should be more appropriate to rephrase the beginning of the paragraph. Instead of “The HIV envelope (Env) trimer is the target for bnAbs, and has some relatively conserved regions, including CD4 binding…” you can use “The bnAbs mainly target some relatively conserved regions on the HIV envelope (Env) trimer, including ….”

Line 105: The author should be explain in brief the  term “SOSIP vaccines”

Line 247:  Remove “in the gastrointestinal tissue” because of redundancy

Line 273:  Remove “vaccinia”

Author Response

The review has been substantially improved after an extensive revision and inclusion of updated references.

Answer: Thanks for your kindly comments.

Minor points:

1) Line 91-97: The authors are mentioning examples of bnAbs targeting different vulnerability sites on the Env trimer. It should be more appropriate to rephrase the beginning of the paragraph. Instead of “The HIV envelope (Env) trimer is the target for bnAbs, and has some relatively conserved regions, including CD4 binding…” you can use “The bnAbs mainly target some relatively conserved regions on the HIV envelope (Env) trimer, including ….”

Answer: Thank you, and we have revised this sentence according to your kind suggestion.

2) Line 105: The author should be explained in brief the term “SOSIP vaccines”

Answer: Thanks for your suggestion, and we have explained it in our revised manuscript. Some works had shown that SOSIP vaccine, a soluble recombinant HIV-1 envelope glycoprotein trimer with natural conformation, elicited robust autologous nAbs in rabbits and non-human primates (NHPs).

3) Line 247: Remove “in the gastrointestinal tissue” because of redundancy

Answer: Thanks for your mention, and we have deleted these abundant words in revised manuscript.

4) Line 273: Remove “vaccinia”

Answer: Thanks for your careful reading, and we have removed it in revised manuscript.